# Detection of Pirimiphos-Methyl in Wheat Using Surface-Enhanced Raman Spectroscopy and Chemometric Methods

**DOI:** 10.3390/molecules24091691

**Published:** 2019-04-30

**Authors:** Shizhuang Weng, Shuan Yu, Ronglu Dong, Jinling Zhao, Dong Liang

**Affiliations:** 1National Engineering Research Center for Agro-Ecological Big Data Analysis & Application, Anhui University, 111 Jiulong Road, Hefei 230601, China; yushuan_1994@163.com (S.Y.); apcomm_2010@163.com (J.Z.); comm_2006@foxmail.com (D.L.); 2Hefei Institute of Physical Science, Chinese Academy of Sciences, 350 Shushanhu Road, Hefei 230031, China

**Keywords:** pirimiphos-methyl, surface-enhanced Raman spectroscopy, wheat, chemometric methods

## Abstract

Pesticide residue detection is a hot issue in the quality and safety of agricultural grains. A novel method for accurate detection of pirimiphos-methyl residues in wheat was developed using surface-enhanced Raman spectroscopy (SERS) and chemometric methods. A simple pretreatment method was conducted to extract pirimiphos-methyl residue from wheat samples, and highly effective gold nanorods were prepared for SERS measurement. Raman peaks assignment was calculated using density functional theory. The Raman signal of pirimiphos-methyl can be detected when the concentrations of residue in wheat extraction solution and contaminated wheat is as low as 0.2 mg/L and 0.25 mg/L, respectively. Quantification of pirimiphos-methyl was performed by applying regression models developed by partial least squares regression, support vector machine regression and random forest with principal component analysis using different preprocessed methods. As for the contaminated wheat samples, the relative deviation between gas chromatography-mass spectrometry value and predicted value is in the range of 0.10%–6.63%, and predicted recovery is 94.12%–106.63%, ranging from 23.93 mg/L to 0.25 mg/L. Results demonstrated that the proposed SERS method is an effective and efficient analytical tool for detecting pirimiphos-methyl in wheat with high accuracy and excellent sensitivity.

## 1. Introduction

Pirimiphos-methyl is a rapid-acting organophosphorus pesticide that is often used for prevention and control of beetles, snout beetles, moths and *Ephestia cautella* during storage of agricultural grains [1]. However, the residue in grains resulting from excessive application is a health hazard for humans and animals [2]. Accurate detection of pirimiphos-methyl residue in grains is crucial to prevent its adverse effects. Gas chromatography-mass spectrometry (GC-MS) [3,4] and liquid chromatography-mass spectrometry [5,6] are the common techniques used for the accurate detection of pesticide residues in grains. However, these techniques have many disadvantages, such as complicated sample pretreatment, large laboratory instruments, and the requirement for trained operating personnel [7,8,9]. Therefore, a method employing portable equipment and a simple detection procedure would be more suitable for detecting pirimiphos-methyl residues in grains.

Surface-enhanced Raman spectroscopy (SERS) is a vibrational spectroscopy technique with high sensitivity; it can provide comprehensive and fingerprint information about analytes without interference from an aqueous phase [10,11,12], which makes it suitable for detecting trace residues in agricultural grains. Given its high sensitivity, specificity and rapidity, SERS has been broadly applied in the detection of various pesticides and toxins, such as aflatoxins [7], carbaryl [9], chlorpyriphos [13], isofenphos-methyl [14] and fenthion [15]. In view of these results, we aimed to detect pirimiphos-methyl residue in wheat using SERS. Spectral measurements are generally executed on a laboratory Raman spectrometer, which is inapplicable in fast detection given its instrument size and rigorous measurement conditions. In the present study, the spectra of pirimiphos-methyl residue were obtained using a portable Raman spectrometer. The signal quality of SERS largely depended on the high-quality substrates [9,16], so the substrate morphology was precisely tuned to obtain a better enhancement effect. In general, acquisition of analyte information from the obtained spectra requires the intervention of professionals. This process is unsuitable for popularization and application of the SERS technique because it is time consuming, laborious and individualistic. However, intelligent spectral analysis based on chemometric methods can eliminate the abovementioned limitations. Commonly used methods, such as partial least squares regression (PLSR) [7,14], artificial neural network [17], support vector machine regression (SVR) [18,19] and random forest (RF) [20,21], are often adopted to develop regression models for quantitative determination of analytes with good predictability. As spectral data are of high dimensionality and contain useless information, many feature extraction methods as non-negative matrix factorization [22] and principal component analysis (PCA) [23,24] are applied to extract the main information and eliminate irrelevant information. In addition, many preprocessing methods as Savitzky−Golay derivation [25,26], wavelet transform [27] and polynomial fitting [28] are commonly used to eliminate background noise and baseline drift. This study aimed to explore the feasibility of applying SERS and chemometric methods to analyze pirimiphos-methyl in wheat using a portable Raman spectrometer. First, a simple pretreatment method was developed to extract pirimiphos-methyl residue in wheat samples. Uniform and efficient gold nanorods (GNRs) were prepared for SERS measurement. Density functional theory (DFT) was used to calculate theoretical Raman spectrum of pirimiphos-methyl and for assignment of characteristic peaks. Then, accurate and quantitative analysis of pirimiphos-methyl was performed by applying regression model developed by PLSR, SVR and RF with PCA using different preprocessed methods as first and second derivative. Contaminated wheat samples that underwent same the pretreatment were also detected to validate the results. 

## 2. Results and Discussion

### 2.1. SERS Substrate

mPEG-SH-coated GNRs were selected as Raman active substrate. From Figure 1, GNRs exhibit two localized surface plasmon resonance (LSPR) bands that are located at 514 and 650-828 nm, and these bands correspond to electron oscillations along the transversal and longitudinal axes of nanorods [8]. 

The length of nanorods can be tuned by varying the silver ion content of the growth solution during synthesis. Resonance will occur when LSPR bands of substrate match laser wavelength of spectrometer. Then, a stronger analyte Raman signal can be obtained. In particular, the longitudinal LSPR band of GNRs varied from 650 nm to 828 nm as AgNO_3_ (0.008 M) content in growth solution was increased from 80 µL to 130 µL. Therefore, mPEG-SH-coated GNRs with LSPR band at 785 nm (AgNO_3_ amount: 120 µL) were prepared for SERS measurement. Meanwhile, the SEM image reveals that the GNRs exhibit high uniformity, which can provide a stable and reliable signal enhancement.

### 2.2. SERS Spectra of Pirimiphos-Methyl

Considering the fingerprint characteristics of SERS, the characteristic spectroscopy peaks reflect molecular vibration and rotation of analytes, which serve as basis for detection. First, DFT was used to calculate the Raman spectrum (RS) of pirimiphos-methyl at B3LYP/6-31G(d) level (Figure 2A). From Raman spectrum of pirimiphos-methyl powder (Figure 2B(a)), the peaks at 574, 631, 653, 829, 854,932, 959, 991, 1076, 1338, 1372, 1439, 1596 and 1635 cm^−1^ are easily observed. According to molecular structure of pirimiphos-methyl and Raman peak assignment (Table 1), the peaks at 574, 653, 829, 854, 959, 992, 1339, 1596, 1636, 1372 and 1439cm^−1^ are assigned to stretching vibration of pyrimidine ring. Stretching vibration of C−C can be associated with 932 cm^−1^. Peaks of 631, 1076 and 1515 cm^−1^ are attributed to stretching vibration of P=S and P−O−CH_3_, wagging vibration of CH_2_ and formation vibration of CH_3_, respectively [29]. 

Given that the differences between the RS and SERS of the analyte, the SERS spectra of pirimiphos-methyl and mPEG-SH-coated GNRs were also measured. As seen in Figure 2B(b,c), peaks of pirimiphos-methyl at 574, 631, 653, 959, 991, 1076, 1338, 1372, 1596 and 1635 cm^−1^ are still obvious. The peak at 961 cm^−1^ of GNRs is close to 959 cm^−1^, and the peaks of 829 and 854 cm^−1^ are also influenced by GNRs. These three peaks can be not used as the discriminating features for the pirimiphos-methyl detection. In addition, considering that the spectra can be influenced by impurities in wheat extraction with making many characteristic peaks change, SERS spectra of wheat extraction solution (Figure 2B(d)) and wheat extraction solution with 25 mg/L pirimiphos-methyl (Figure 2B(e)) were collected. As for wheat extraction solution with 25 mg/L pirimiphos-methyl, the characteristic peaks at 574, 631, 991, 1076 and 1372 cm^−1^ are visible and unaffected, but peaks at 653, 1596, 1338 and 1635 cm^−1^ are overlapped with peaks of blank wheat extraction solution. Therefore, these results demonstrate that SERS has preliminary feasibility for detection of pirimiphos-methyl in wheat extract solutions. 

Then, spectra of wheat extract solution with 25, 10, 5, 2.5, 1, 0.5, 0.2 and 0.1 mg/L pirimiphos-methyl were measured using SERS with mPEG-SH-coated GNRs (Figure 3). The spectra in Figure 3 were initially baseline-corrected, and the spectra were shifted vertically for better presentation. From the figure, intensity of peaks at 574, 631, 991, 1076 and 1372 cm^−1^ gradually decreases with decreasing residue concentration. However, when residue concentration is below 0.2 mg/L (0.1mg/L), most of characteristic peaks are no longer observed, which suggests that pirimiphos-methyl in wheat extract solution of 0.2 mg/L can be detected using SERS with GNRs on a portable Raman spectrometer.

In addition, spectra repeatability is vital for an accurate determination. The intensity variation of characteristic peaks at 574, 631, 991, 1076 and 1372 cm^−1^ from 20 samples containing 2.5 mg/L pirimiphos-methyl was shown in Figure 4. 

As seen in the figure, spectral peaks of different samples have high repeatability and exhibit small variation, and relative standard deviation (RSD) is only 8.31%, which provides reliable SERS detection. It is noticed that the intensity of some specific peaks of one sample showed different variation trends compared with other samples. It is mainly due to the fact that the “hot spots” have different enhanced effect on same band in different sample, which may lead to different intensity ratios for two specific bands in different samples.

### 2.3. Comparison of Spectral Analysis Using Different Chemometric Methods

SERS spectra were preprocessed by a Savitzky−Golay function to obtain first and second derivatives of spectra, which eliminated baseline and linear slope effects. The original spectra, the first derivative of spectra and the second derivative of spectra were used for the subsequent analysis. Then, quantification of pirimiphos-methyl in wheat was performed by applying regression models developed using PLSR, SVR and RF, and PCA was used for feature extraction. Analysis results of regression models were shown in Table 2. The predicted error of the RF models is high in some cases. Predicted performance of models developed using PLSR and different preprocessed spectra is similar, and application of PCA can obtain the better model. RMSEC and RMSEP of optimal PLSR model are 0.0051 mg/L and 0.0096 mg/L, respectively. SVR models always obtained good prediction results, and the lowest RMSEP is 0.0147 mg/L. By contrast, the best model was constructed using PLSR and PCA with original spectra, and the predicted results for wheat extraction solution with residue of different concentration were shown in Figure 5. As seen in the figure, the model can predict the concentration of pirimiphos-methyl in wheat solutions of the calibration and validation sets with low error, thus the model was adopted for quantitative analysis of pirimiphos-methyl in wheat samples in subsequent analysis.

### 2.4. Quantification of Pirimiphos-Methyl Residue in Wheat

Wheat samples contaminated with pirimiphos-methyl were extracted using the presented pretreatment method. Then, the obtained extract solutions were used for SERS measurement. The spectra of ten representative samples are shown in Figure 6. 

The spectra in Figure 6 were initially baseline-corrected, and the spectra were shifted vertically for better presentation. From the figure, the spectra of extract solutions of pirimiphos-methyl residue in wheat are highly consistent with the spectra of wheat extract solutions containing pirimiphos-methyl, which proves the feasibility for prediction of residue in wheat using the above established model. Meanwhile, the lowest tested concentration of 0.25 mg/L is far below maximum residue limit of pirimiphos-methyl in wheat (5 mg/L). Then spectra were processed using PCA, and the obtained principal component scores were used to predict pirimiphos-methyl concentration based on the established models. Actual residue values for contaminated wheat samples were measured using GC-MS. Comparing actual value with predicted value (Table 3), the relative deviation is in the range of 0.10% to 6.63%, and the predicted recovery is from 94.12% to 106.63%. 

Figure 7 shows the values measured by GC-MS and SERS are basically consistent with actual value directly. The results also indicate the pretreatment method in this study is feasible and effective for the extraction of pirimiphos-methyl residue in wheat. Meanwhile, the predicted standard deviation is from 0.010 mg/L to 0.112 mg/L, which demonstrates that SERS can provide stable detection. Accordingly, SERS with mPEG-SH-coated GNRs, PLSR and PCA can detect pirimiphos-methyl residues in wheat with high sensitivity and good repeatability when coupled with the presented extraction method. Method detection limit (MDL) and reliable quantitation limit (RQL) were used to assess the limit-of-detection of the presented methodology [30]. MDL of pirimiphos-methyl residue in wheat was 0.0442 mg/L, and RQL was 0.1768 mg/L.

## 3. Materials and Methods

### 3.1. Reagents

Acetone, hydrogen tetrachloroaurate (HAuCl_4_·3H_2_O), trisodium citrate, L-ascorbic acid, sodium borohydride (NaBH_4_), silver nitrite (AgNO_3_) and methoxymercaptopoly(ethylene glycol) (mPEG-SH) were obtained from Aladdin Reagent Co., Ltd. (Shanghai, China). Pure pirimiphos-methyl powder (99.8%) was purchased from Beijing Tanmo Technology Co., Ltd. (Beijing, China). Wheat samples were obtained from Hefei Zhougudui Agriculture Products Wholesale Market (Hefei, China).

### 3.2. DFT Calculation

Geometry optimization and vibrational spectra (including the Raman spectrum) for pirimiphos-methyl were calculated using DFT in Gaussian 09w program. For DFT calculations [31,32], Beckes three-parameter hybrid exchange function (B3) and the correlation function of Lee, Yang and Parr (LYP) were adopted, and 6-311G(d) was used as the basis set.

### 3.3. Sample Preparation

Extraction method for wheat samples was developed on the basis of sample preparation protocol used in GC detection (SNT 2324-2009). Wheat samples were homogenized with a pulverizer and filtered through 10 mesh sieves. A total of 5.00 g wheat powder was mixed with 5 mL of deionized water and 15 mL of acetone in 50 mL graduated centrifuge tube. Then, the mixture was placed in an oscillator with shaking extraction for 10 min and centrifuged at 4000 rpm for 5 min. Supernatant was transfered to a 50 mL centrifuge tube. Wheat residual was extracted with 15 mL of acetone again, and supernatant was merged with the previous supernatant. All the supernatant was filtered with 0.22 µm organic filtration and then evaporated to 5 mL in 60 °C water bath. And the condensed supernatant was used for SERS measurement. 

Wheat extraction solutions with pirimiphos-methyl were first prepared as reference solutions of residue. Briefly, pure pirimiphos-methyl powder was diluted in the condensed supernatant of uncontaminated wheat to get different concentrations: 25, 10, 5, 2.5, 1, 0.5, 0.2 and 0.1 mg/L. 

Forty contaminated wheat samples were obtained from Center of Agricultural Products Quality and Safety, Anhui Academy of Agricultural Sciences. Actual values of 40 samples were obtained using a GC-MS instrument (TSQ8000EVO, Thermo Fisher, Waltham, MA, USA), and detection procedure was performed according to Huang’s work [13]. GC-MS results were provided by Center of Modern Experimental Technology, Anhui University. Pirimiphos-methyl residues in the wheat samples ranged from 23.93 mg/L to 0.25 mg/L. 

### 3.4. SERS Measurement

The mPEG-SH-coated GNRs were prepared and selected as the SERS substrate. GNRs were synthesized using a seed-mediated growth method [33], and mPEG-SH was adopted to displace cetyltrimethylammonium bromide from the GNRs surface. The assembly method we adopted was originally proposed by Zhou [34]. During SERS measurement, mPEG-SH can induce self-assembly and prevent the aggregation of GNRs, which provides consistent and efficient enhancement. Resonance can occur when the plasmon resonance bands of substrate match the laser wavelength of spectrometer. Then, a stronger analyte Raman signal will be obtained. Therefore, plasmon resonance bands of GNRs were tuned by varying the silver ion content of the growth solution during synthetic process in this study. GNRs morphologies were measured using a scanning electron microscope (SEM, Hitachi S4700, Tokyo, Japan) and a UV-2600 ultraviolet-visible (UV–Vis) spectrometer (Shimadzu, Kyoto, Japan).

GNRs sol-solution was centrifuged at 8500 rpm for 10 min to obtain the gray colloid. And 2 μL of GNRs colloid was dropped on a silicon chip. Afterward, 2 μL of testing solution was added onto the dried GNRs film. SERS spectra of the dried droplet were obtained on a portable Raman spectrometer (B&WTEK, i-Raman785^®^ Plus, Newark, DE, USA) with a 785 nm laser of 150 mW. The spectral resolution was about 3.5 cm^−1^, and the detector was thin backlit CCD array. The diameter of sampled region was about 100 μm. Integration time was 5 s, and spectra were recorded with three scans in Raman shift of 550 to 1750 cm^−1^. For pure pirimiphos-methyl, the pirimiphos-methyl powder was placed on a silicon chip, and then Raman spectra were obtained. And acquirement of all the spectra in this study was at same measurement conditions. 

As for wheat extraction solution with residue, twenty samples were prepared for each concentration. Five spectra were measured as the representative spectra from five different points for each sample. Forty contaminated wheat samples that underwent the above pretreatment procedures were used for SERS measurement, and five spectra were also collected for each sample.

### 3.5. Data Analysis

Spectra were initially baseline-corrected and normalized to minimize the influences from instruments and measuring environment. The corrected SERS spectra were preprocessed using a Savitzky−Golay function to obtain first and second derivatives of spectra, which can eliminate baseline and linear slope effects. Then, quantification of pirimiphos-methyl in wheat was performed by applying regression models developed using PLSR, SVR and RF. PCA was used for feature extraction of spectra. Prior to application of chemometrics methods, all the spectral data were divided into a calibration set and validation set (4:1) by a Kennard-Stone algorithm. The calibration set was for developing regression models, and the validation set was for testing the models. Prediction performance of model was quantitatively evaluated coupling with root-mean-square error of calibration (RMSEC) and root mean standard error of prediction (RMSEP). All data analyses were performed in MATLAB 2013a (The MathWorks Inc., Natick, MA, USA). 

## 4. Conclusions

In this study, a novel method was developed for the detection of pirimiphos-methyl residues in wheat using SERS with chemometric methods on a portable Raman spectrometer. The mPEG-SH-coated GNRs were prepared for SERS measurement, and a simple pretreatment method was developed for extracting residue in wheat. Pirimiphos-methyl values of 0.2 mg/L and 0.25 mg/L in wheat extract solution and wheat samples can be detected, which are far below the maximum residue limit of pirimiphos-methyl of China. Values of pirimiphos-methyl residue were predicted by applying regression models developed by PLSR, SVR and RF with PCA using different preprocessed methods. Comparing GC-MS value with predicted value for contaminated wheat samples, relative deviation is in the range of 0.10 %–6.63%, and predicted recovery are from 94.12 % to 106.63 %. These results indicated that the presented method is an effective and feasible approach for the determination of pirimiphos-methyl residue in wheat. For detection of residue in grains, a suitable pretreatment is essential and crucial. Spectral variation induced by the instability of nanoparticles and differences in sampling and sample batches should be minimized and avoided prior to SERS measurement. Meanwhile, it is useful to develop intelligent spectral processing and statistical analysis for SERS detection at critical locations for the safety and quality of grains. Accordingly, SERS is a promising and potentially powerful tool for detecting pirimiphos-methyl or other pesticides and toxic residues in grains. 

## Figures and Tables

**Figure 1 molecules-24-01691-f001:**
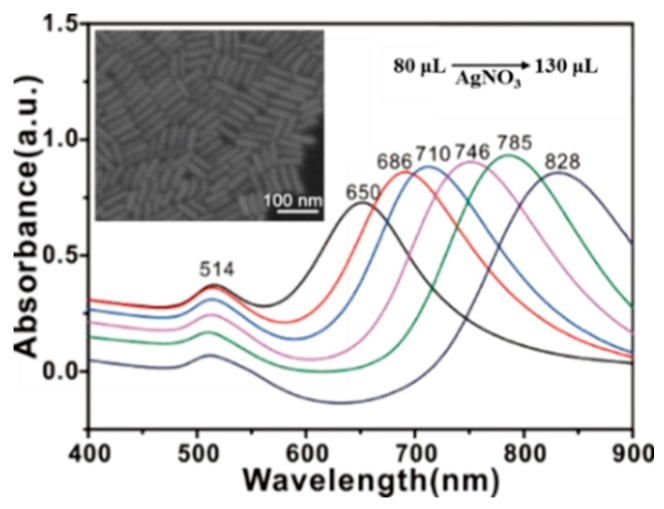
UV–visible spectra of GNRs solutions with addition of AgNO_3_ (0.008 M) at 80, 90, 100, 110, 120 and 130 µL in growth solution. The inset shows a SEM image of GNRs with a longitudinal LSPR band at 785 nm.

**Figure 2 molecules-24-01691-f002:**
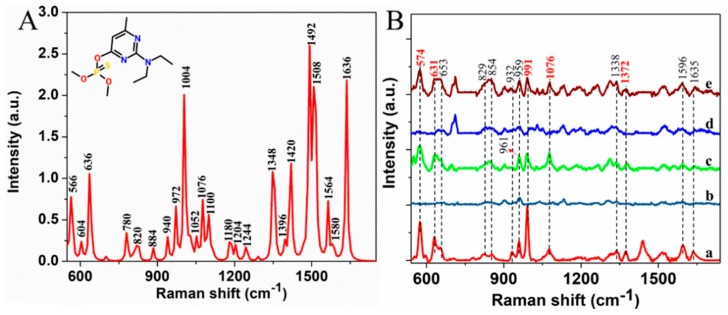
(**A**): Calculated Raman spectra of pirimiphos-methyl using DFT, and the inset is molecular structure of pirimiphos-methyl; (**B**): Raman spectra of pirimiphos-methyl powder (**a**) and SERS spectra of acetone (**b**), pirimiphos-methyl in acetone solution (25 mg/L) (**c**), wheat extraction solutions (**d**) and pirimiphos-methyl in wheat extraction solutions (25 mg/L) (**e**).

**Figure 3 molecules-24-01691-f003:**
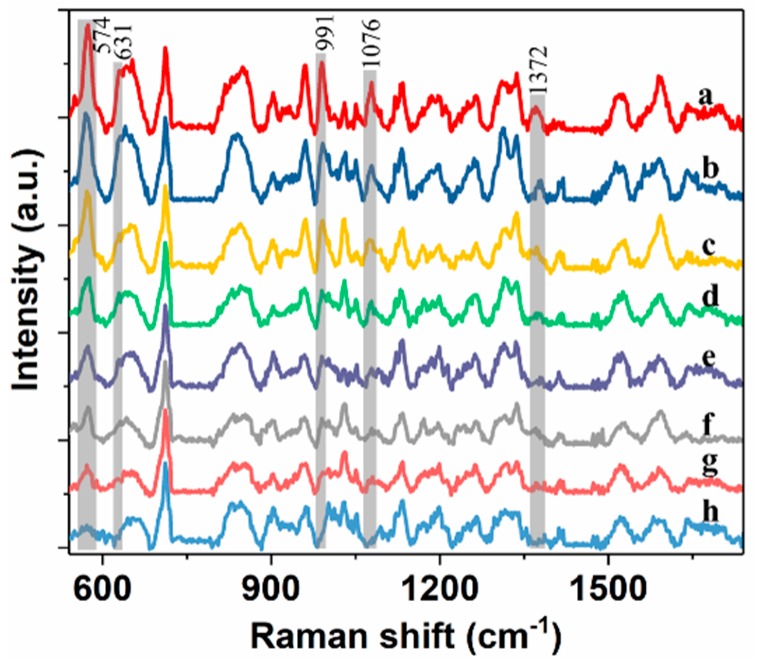
SERS spectra of wheat extract solution with pirimiphos-methyl residue of 25, 10, 5, 2.5, 1, 0.5, 0.2 and 0.1 mg/L (**a**–**h**).

**Figure 4 molecules-24-01691-f004:**
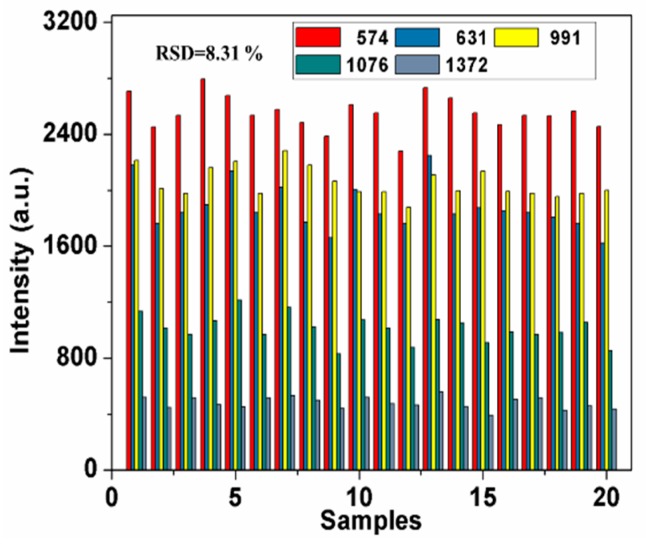
Intensity variation of peaks at 574, 631, 991, 1076 and 1372 cm^−1^ from twenty samples containing 2.5 mg/L pirimiphos-methyl.

**Figure 5 molecules-24-01691-f005:**
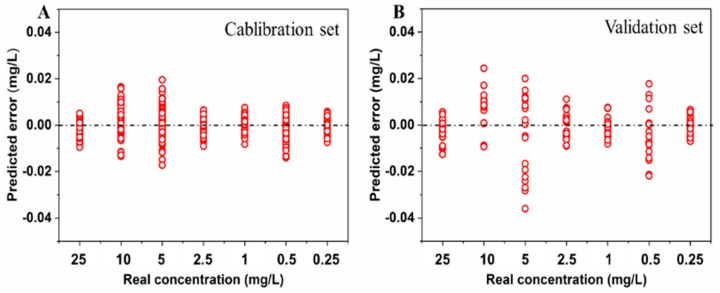
Predicted error of the optimal regression model for wheat extraction solution with residue of different concentration ((**A**): calibration set, (**B**): validation set).

**Figure 6 molecules-24-01691-f006:**
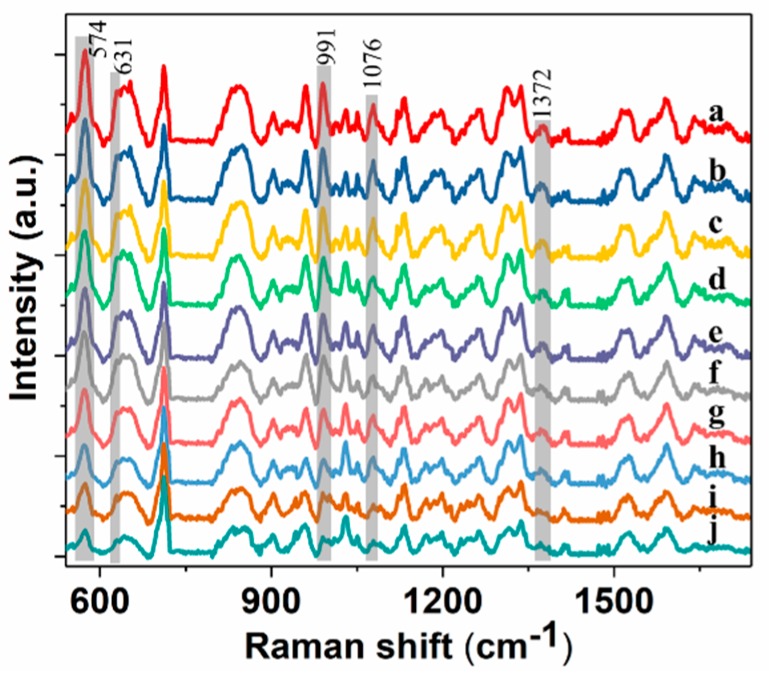
Spectra of pirimiphos-methyl residues in the wheat samples of 23.93, 15.85, 11.72, 9.05, 7.36, 4.75, 3.49, 1.45, 0.91 and 0.25 mg/L (**a**–**j**) with the proposed extraction method.

**Figure 7 molecules-24-01691-f007:**
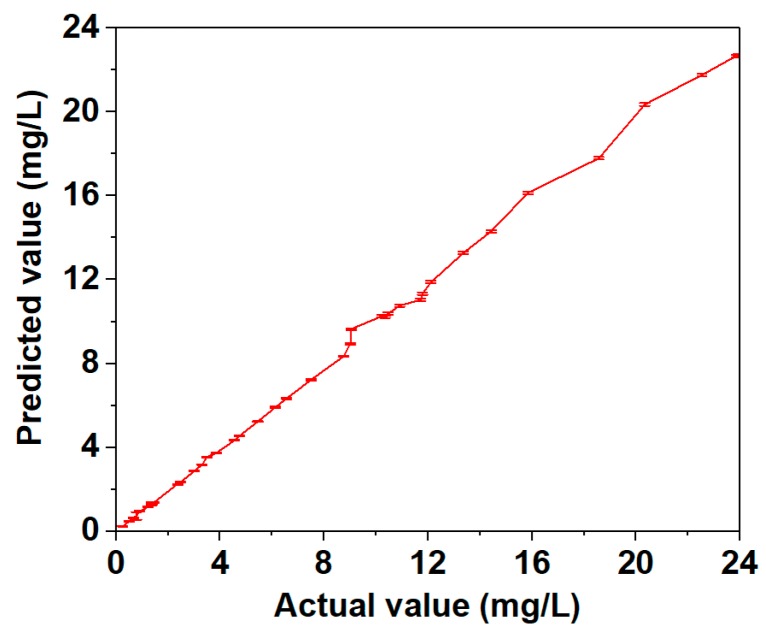
Comparison of the values measured by GC-MS (actual value) and SERS (predicted value).

**Table 1 molecules-24-01691-t001:** Peak assignment for calculated and experimental Raman peaks of pirimiphos-methyl (cm^−1^).

Calculated Raman Peaks	Experimental Raman Peaks	Assignment
566, 604, 780, 820, 884, 972, 1004, 1100, 1180,1348, 1564, 1580, 1636, 1396, 1420	574, 653, 829, 854, 959, 992, 1339, 1596, 1635, 1372, 1439	stretching vibration of pyrimidine ring
636	631	stretching vibration of P=S
940	932	stretching vibration of C−C
1052		stretching vibration of P−O−CH_3_
1076	1076	stretching vibration of P−O−CH_3_
1204		formation vibration of CH_3_
1244		asymmetrical stretching vibration of N−CH_2_
1492		stretching vibration of C−N
1508	1515	formation vibration of CH_3_

**Table 2 molecules-24-01691-t002:** Predicted results of regression models developed with three preprocessed spectra for pirimiphos-methyl concentrations in wheat extraction (units: mg/L).

Methods	Number of Latent Variables	Original Spectra		1st Derivative		2nd Derivative	
		RMSEC	RMSEP	RMSEC	RMSEP	RMSEC	RMSEP
PLSR	23	0.3067	0.4012	0.2502	0.3221	0.3333	0.4109
SVR		0.0084	0.0272	0.0086	0.0193	0.0092	0.0359
RF		0.1581	0.1906	0.2192	0.3833	0.3679	0.8653
PCA+PLSR	9	0.0051	0.0096	0.0092	0.0221	0.0053	0.0142
PCA+SVR		0.0094	0.0147	0.0095	0.0157	0.0092	0.0160
PCA+RF		0.6737	1.6748	0.6225	1.6801	0.6410	1.6641

**Table 3 molecules-24-01691-t003:** Predicted results of contaminated wheat with pirimiphos-methyl using SERS, PLSR and PCA.

Actual Value by GC-MS (mg/L)	Predicted Values by SERS		Relative Deviation (%)	Recovery (%)	Error (mg/L)
	Mean Value (mg/L)	Standard Deviation (mg/L)			
23.93	22.71	0.112	5.10	94.90	1.22
23.86	22.67	0.111	4.99	95.01	1.19
22.54	21.76	0.102	3.46	96.54	0.78
20.34	20.36	0.105	0.10	100.10	−0.02
18.57	17.78	0.103	4.25	95.75	0.79
15.85	16.13	0.105	1.77	101.77	−0.28
14.43	14.31	0.103	0.83	99.17	0.12
13.35	13.27	0.099	0.60	99.40	0.08
12.11	11.89	0.097	1.82	98.18	0.22
11.79	11.34	0.095	3.82	96.18	0.45
11.72	11.04	0.093	5.80	94.20	0.68
10.91	10.76	0.091	1.37	98.63	0.15
10.47	10.37	0.090	0.96	99.04	0.1
10.35	10.23	0.087	1.16	98.84	0.12
10.23	10.27	0.079	0.39	100.39	−0.04
9.05	9.65	0.074	6.63	106.63	−0.6
9.03	8.95	0.071	0.89	99.11	0.08
9.01	8.96	0.072	0.55	99.45	0.05
8.76	8.35	0.068	4.68	95.32	0.41
7.50	7.23	0.069	3.60	96.40	0.27
6.56	6.34	0.058	3.35	96.65	0.22
6.13	5.92	0.059	3.43	96.57	0.21
5.45	5.24	0.048	3.85	96.15	0.21
4.75	4.55	0.042	4.21	95.79	0.2
4.56	4.34	0.041	4.82	95.18	0.22
3.87	3.75	0.041	3.10	96.90	0.12
3.49	3.54	0.043	1.43	101.43	−0.05
3.31	3.18	0.042	3.93	96.07	0.13
3.01	2.88	0.039	4.32	95.68	0.13
2.37	2.23	0.037	5.91	94.09	0.14
2.47	2.36	0.036	4.45	95.55	0.11
1.45	1.37	0.033	5.52	94.48	0.08
1.38	1.27	0.029	7.97	92.03	0.11
1.36	1.37	0.031	0.74	100.74	−0.01
1.23	1.17	0.028	4.88	95.12	0.06
0.91	0.96	0.027	5.49	105.49	−0.05
0.79	0.75	0.34	5.06	94.94	0.04
0.67	0.66	0.027	1.49	98.51	0.01
0.51	0.48	0.010	5.88	94.12	0.03
0.25	0.24	0.011	4.00	96.00	0.01

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
