# Peer review of "Detection of Pirimiphos-Methyl in Wheat Using Surface-Enhanced Raman Spectroscopy and Chemometric Methods"

_molecules, 2019, doi:10.3390/molecules24091691_

Round 1
Reviewer 1 Report
The present paper reports on the application of SERS for the detection of pirimiphos-methyl in wheat. In particular, the authors prepare substrates for SERS measurements of pirimiphos-methyl extracted from wheat and adopted different chemometric methods for pesticide quantification. Original results are reported and the topics addressed by the authors is interesting, but the article is written in an approximate way and presents some inconsistencies. Furthermore, many significant details are missing. For example, the authors propose to use SERS and multivariate analysis for the determination of pesticide concentration. Table 3 presents the related actual values, the predicted ones and some statistical parameters evaluated from the numerical values. In this Table, the authors use two different units of measurement that can be considered equivalent only in the case in which the examined solutions are obtained using water. At first glance, this is not the present case since samples investigated by GC-MS method use solvent different from water. In addition, the actual and predicted values are reported without the error that is a crucial information in this case. Many other important aspects need to be significantly improved in order to increase the readability of the paper and strengthen the work.
For these reasons, the paper cannot be accepted in the present version for publication on Molecules but a deep revision is required.
In the following a detailed list of the points that need to be addressed
a) When it is possible please avoid to use different units of measurements for the same quantity.
b) At line 61 the authors report on the different uses of Savitzky-Golay derivative. Please note that the cited reference is not appropriate and specify how the Savitzky-Golay derivative can be used for baseline correction. Add some pertinent references.
c) At line 70 please clarify what the authors mean by “different preprocessed methods” because they just discuss Savitzky-Golay derivative in 3.2 paragraph.
d) Please add details about the spectral resolution of the Raman spectrometer, the size of the sampled region and the detector used.
e) Give also details about the measurement conditions for Raman measurements reported in Fig. 2
f) Please add information about DFT calculations in the Materials &Methods section.
g) Please add some references for the peak assignments reported in Table 1. In order to validate these assignments, it’s fundamental to take into account the spectral resolution of Raman spectrometer.
h) Please rewrite in a clearer way the sentences reported at lines 173-174.
i) The readability of Figs. 3 and 6 will be improved if the Raman shift position of the main peaks is indicated. Moreover, indicate what preprocessing analysis has been performed on the reported spectra.
j) The title of 3.3 paragraph is not correct, the topic of this section is the comparison of different chemometric methods. The considered spectra are the ones reported in Fig.3? Are they already preprocessed before reporting in Fig.3? The discussion here reported is not very clear. Please improve also the caption of Fig. 5.
k) The determination of limit-of-detection is not clear. Please give additional details.
l) Please carefully revise the English language and the Reference section. In some cases, the references are not correctly cited in the manuscript (e.g. see line 140)
Author Response
Thanks for your review. The revised parts in out manuscript have labelled in red.

Reviewer 2 Report
The manuscript "Detection of pirimiphos-methyl in wheat using surface-enhanced Raman spectroscopy and chemometric methods", written by Weng et al., describes the novel method for a determination of pirimiphos-methyl using SERS. The importance of such method is relatively high, and it can find applications in many fields. However, there are still some issues, which need to be answered. The method is based on mPEG-SH-coated GNRs, tuned for the laser irradiation at 785 nm.
1. The spectra present in figure 2B are extremely different, and I am convinced that particular spectral bands present in figure 2Be can’t be interpreted as parts of the analyte. One sample of wheat extract/and one sample of spiked sample do not represent a suitable sample set for analysis. Much more data should be acquired, and average spectra compared.
2. Data present in figure 4 are not convincing. When the intensity of one spectral band is higher for two compared samples (samples 4 and 5 for example, however, there are more), intensity of the rest two bands is lower, which could indicate nonspecific interactions. Can you please comment on this?
3. How was the RSD in the figure 4 calculated? Is it an average from all data or for one spectral band?
Author Response

(The authors gave the same response as above.)

Round 2
Reviewer 1 Report
The authors improved their manuscript but some efforts are still needed in order to make the manuscript worthy of publication on Molecules Journal.
1) I appreciated that the authors introduced some details an references about DFT calculations but I believe that they should not be inserted in the SERS measurement paragraph. Please add a proper small paragraph for DFT calculations
1) Given the spectral resolution of 3.5 cm-1, many assignments reported in Table 1 have to be carefully reconsidered. In many cases the difference between calculated and experimental detected peaks position makes the assignment critical (see for instance peaks generally assigned to stretching vibration of pyrimidine ring, the peaks at 1396 and 1420 cm-1).
2) The sentences (Peak at 961 cm−1of GNRs is close to 959 cm−1, and peak at 959 cm−1of 178 pirimiphos-methyl is obstructed. And the peaks of 829 and 854 cm−1are also influenced by the 179 GNRs.These peaks are unavailable for the pirimiphos-methyl detection) even though has been changed are still not clear and with misprints. Please clarify what the authors mean for obstructed
3) Also the sentences added at 203-207 lines are not clear.
4) As I said in my previous report, please add errors in the data regarding the actual values measured by GC-MS and SERS. Perhaps an additional figure with actual data and SERS data with their error bars would be useful for showing the agreement among the results of the two different measurement techniques.
5) Some typos and English language errors are still present.
Author Response

(The authors gave the same response as above.)

Reviewer 2 Report
The manuscript "Detection of pirimiphos-methyl in wheat using surface-enhanced Raman spectroscopy and chemometric methods" was considerably improved and all my comments were answered. However, I still believe that the new sentence, line 204 - 207 should be re-phrased in a more scientific language and considerably extended. Is there any hypothesis describing this phenomenon?
Author Response

(The authors gave the same response as above.)
